# Distribution of *Corynebacterium* Species and Comparative Results of Diagnostic Methods for Identifying *Corynebacterium* in Experimental Mice in Korea

**DOI:** 10.3390/vetsci9070328

**Published:** 2022-06-29

**Authors:** Sehee Park, Hijo Shin, Sangwoon Kim, Teakchang Lee, Haejin Lee, Kihoan Nam, Wonkee Yoon, Hyoungchin Kim, Youngwon Seo, Youngsuk Won, Hyojung Kwon

**Affiliations:** 1Laboratory Animal Resource Center, Korea Research Institute of Biology and Biotechnology, 30, Yeongudanji-ro, Cheongwon-gu, Cheongju-si 28116, Korea; seheepark@kribb.re.kr (S.P.); yoyoyep@kribb.re.kr (H.S.); z5r5@kribb.re.kr (S.K.); hp6842@kribb.re.kr (T.L.); z6p9@kribb.re.kr (H.L.); namk@kribb.re.kr (K.N.); wkyoon@kribb.re.kr (W.Y.); hckim@kribb.re.kr (H.K.); forever@kribb.re.kr (Y.S.); 2Department of Veterinary Pathology, College of Veterinary Medicine, Chungnam National University, 99, Daehak-ro, Yuseong-gu, Daejeon 34134, Korea

**Keywords:** *Corynebacterium*, laboratory mice, MALDI-TOF MS

## Abstract

**Simple Summary:**

Experimental mice are the most commonly used laboratory animals for biomedical research and comparative studies. However, microbial infection may alter the mouse phenotype and confound interpretation results. The genus *Corynebacterium*, Gram-positive diphtheroid rod-shaped bacteria, induces severe diseases, such as hyperkeratosis and pseudotuberculosis, in immunodeficient mice. In this report, we described the population of *Corynebacterium* spp. isolated from laboratory mice in Korea using different approaches, comparing the accuracy and problems associated with each method. When identified based on molecular methods such as 16S *rRNA* and *rpoB* gene sequence analysis, the main *Corynebacterium* species were *C. mastitidis* (44.8%), *C. bovis* (25.5%), *C. lowii* (21.2%), and *C. amycolatum* (8.5%). In addition, matrix-assisted laser desorption/ionization–time of flight mass spectrometry (MALDI-TOF MS) yielded results that were 77.9% identical to the molecular results, whereas biochemical methods showed only 15.5% identical to molecular identification. Collectively, our findings indicate that the different results may be obtained depending on the method used to identify *Corynebacterium* isolated from experimental mice, highlighting the importance of selecting an appropriate *Corynebacterium* identification method in obtaining accurate identification results. This result will help to increase the reliability of *Corynebacterium* diagnosis result from experimental mice.

**Abstract:**

The genus *Corynebacterium*, composed of Gram-positive diphtheroid rod-shaped bacteria, induces severe diseases, such as *Corynebacterium*-associated hyperkeratosis and pseudotuberculosis, in immunodeficient mice. We isolated and identified a total of 165 strains of *Corynebacterium* species from experimental mice in Korean laboratories, diagnosed using several methods. When identified based on molecular methods, namely, 16S *rRNA* and *rpoB* gene sequence analysis, the main *Corynebacterium* species isolated in Korean laboratory mice were *C. mastitidis* (44.8%, *n* = 74), *C. bovis* (25.5%, *n* = 42), *C. lowii* (21.2%, *n* = 35), and *C. amycolatum* (8.5%, *n* = 14). Diagnoses were also performed using matrix-assisted laser desorption/ionization–time of flight mass spectrometry (MALDI-TOF MS) and biochemical methods. MALDI-TOF MS yielded results that were 77.9% identical to the molecular identification results, whereas biochemical methods showed only 15.5% identical to molecular identification, partly owing to difficulties in distinguishing among *C. mastitidis* strains. Collectively, our findings indicate that molecular biological methods are better suited for detecting and identifying *Corynebacterium* species candidates isolated from mice than biochemical methods. Because of limitations associated with the use of MALDI-TOF MS, more precise results will be obtained by complementing this approach with other methods when used for rapid identification testing.

## 1. Introduction

Bacteria of the genus *Corynebacterium* are Gram-positive, short coryneform (club-shaped) rods that are catalase-positive and generally aerobic [1]. As of today, the genus *Corynebacterium* contains 133 validated species that are widely distributed in nature and occasionally are pathogens of animals [2,3].

Experimental mice are the most commonly used laboratory animals for biomedical research and comparative studies. However, microbial infection may alter the mouse phenotype and confound interpretation results. Therefore, confirming that animals are pathogen-free through regular and repeated tests of animal colonies for selected pathogens is important to obtain reproducible and reliable animal test results. Such microbiological monitoring programs are essential to the operation of experimental animal facilities, not only to guarantee quality experimental results, but also for general animal welfare and to protect researchers against zoonotic disease [4].

Although the distribution of *Corynebacterium* spp. in experimental mice and rats is not well described, some studies on pathogenic *Corynebacterium* have been reported. For example, *C. kutscheri* can cause pseudotuberculosis, characterized by rough hair coat, slowed activity, leanness, and hyperpnea [5]. In addition, *Corynebacterium*-associated hyperkeratosis (CAH), caused by *C. bovis*, has been the subject of reports in recent decades [6]. Also known as “scaly skin disease”, CAH has been reported in South Korea [7]. Therefore, these two *Corynebacterium* were included among the regular microbiological monitoring test items and evaluated in many animal facilities, including in Korea [8]. Federation of European Laboratory Animal Science Associations (FELASA) recommends these pathogenic bacteria as a regular health monitoring item [4]. Therefore, accurate diagnosis of *Corynebacterium* in laboratory mice by a reliable method is very important.

*Corynebacterium* is usually diagnosed through cultivation and biochemical phenotypic analysis of isolates using an analytical profiling index system (API Coryne; bioMérieux, Marcy-l’Étoile, France) [9,10]. *Corynebacterium* in human isolates is correctly detected in only 25% to 30% of cases using this approach [11], highlighting limitations in diagnosing *Corynebacterium* based solely on biochemical phenotype. However, diagnosing and detecting *Corynebacterium* in laboratory animals using API Coryne system have not been reported yet. 

One approach for resolving these limitations is to use sequencing of 16S *rRNA* genes as a standard for determining bacterial species, based on the concept that the regularity of mutations in these genes over time constitutes a type of “molecular clock” [12]. However, the 16S *rRNA* gene also has a drawback in that it is not polymorphic enough to ensure reliable phylogenetic differentiation [13]. Sequencing of the RNA polymerase beta subunit-encoding gene (*rpoB*) was proposed as an improvement compared with identification based on the 16S *rRNA* gene sequence. This method was able to identify *Corynebacterium* spp. with high accuracy, even using short sequences of 434 to 452 base pairs [14].

Previously, matrix-assisted laser desorption ionization–time of flight mass spectrometry (MALDI-TOF MS) has been used for clinical microbial diagnosis [15], with several papers highlighting its low cost, speed, and accurate identification of pathogenic *Corynebacterium* isolated from humans. However, there have been no reports of diagnosis of *Corynebacterium* infection in mice using MALDI-TOF MS.

In this report, we describe the population of *Corynebacterium* spp. isolated from laboratory mice in Korea using different approaches, comparing the accuracy and problems associated with each method.

## 2. Materials and Methods

### 2.1. Ethical Consideration

The study protocol was approved by and conformed to the guidelines of the international animal care and use committee (IACUC) of the Korea Research Institute of Bioscience and Biotechnology.

### 2.2. Sample Collection

Ninety-three mice infected with *Corynebacterium* spp. were obtained from five research institutes, four companies, 11 universities, and two hospitals in Korea. Animals were sacrificed by exsanguination under deep isoflurane anesthesia. *Corynebacterium* spp. were isolated by performing culture tests as described in our previous report [16]. Briefly, skin, nasal cavity, and trachea were wiped with a moistened cotton swab and streaked onto trypticase soy agar containing 5% sheep blood (BD Biosciences, San Jose, CA, USA). After incubating aerobically for 48, 72, or 96 h at 37 °C, small and dome-shaped nonhemolytic candidate bacteria colonies were recovered and initially screened using a combination of Gram-staining, colony morphology assessment, and catalase positivity. Samples were collected from 2015 to 2021.

### 2.3. Identification by Biochemical Tests

Tests were carried out using the API Coryne system (bioMérieux, Marcy-l’Étoile, France, ver. 4.0, https://apiweb.biomerieux.com accessed on 28 June 2022) as described by the manufacturer. Briefly, each inoculum was prepared from a heavy suspension in distilled water having an opacity >6 on the McFarland scale. Each inoculum was subjected to enzymatic and fermentation tests. Strips were incubated at 37 °C for 24 or 48 h. Test results were read by two researchers; in the event of a discrepancy, a third person judged the results. Identification is based on a calculation of how closely the profile corresponds to the taxon relative to all other taxa in the database. As per the manufacturer’s recommendation, species identification was supported in cases where the correspondence was greater than 80%.

### 2.4. Identification by MALDI-TOF MS

MALDI-TOF testing was performed using a MALDI Biotyper (MBT ver. 4.1.100, Billerica, MA, USA) system, which can detect 70 species and 288 strains of *Corynebacterium*, and an MBT Compass reference library ver. 4.1 (Bruker Daltonics, Billerica, MA, USA), as described by the manufacturer. Briefly, colonies were transferred to a metallic MALDI-TOF MSP 96 plate (MSP 96 target polished steel BC; Bruker Daltonik, Billerica, MA, USA), after which 1 μL of matrix (α-cyano-4-hydroxycinnamic acid (HCCA), 50% acetonitrile, 2.5% trifluoroacetone) was applied dropwise. As per the manufacturer’s description, if a score of 2.0 or higher is obtained, identifications are judged to be properly obtained on species level. Each sample was tested at least twice. In cases where valid results could not be obtained, an ethanol/formic acid extraction was performed.

### 2.5. Molecular Identification and Sequence Analysis

Molecular identification was performed by sequencing the 16S *rRNA* gene and sequencing of the *rpoB* gene, as previously described [13,14]. Briefly, polymerase chain reaction (PCR) was performed using universal primer pairs for each gene, as follows: 16S *rRNA*, 5′-TAC CTT GTT ACG ACT T-3′ (1492R primer) and 5′-AGA GTT TGA TCM TGG CTC AG-3′ (27F primer) [17]; and *rpoB,* 5′-CGW ATG AAC ATY GGB CAG GT-3′ (C2700F primer) and 5′-TCC ATY TCR CCR AAR CGC TG-3′ (C3130R primer) [13]. PCR-amplified DNA fragments were sequenced and compared to the GeneBank database (National Center for Biotechnology Information) using BlastN [18,19,20]. *rpoB*-based identification to the species level required ≥95% sequence identity with ≥2% separation between species, whereas 16S *rRNA* gene-based identification to the species level required ≥99% identity with ≥0.8% separation between species.

### 2.6. Histopathological Analysis

In cases where visual lesions were found in mice, tissues were fixed with 10% neutralized formalin (Sigma-Aldrich, St. Louis, MO, USA) for 48 h. The tissue samples were subsequently processed in an automatic tissue processor (Tissue-Tek, Sakura, Tokyo, Japan), embedded in paraffin, sectioned, and stained with standard hematoxylin and eosin (H&E; Abcam, Cambridge, UK) for evaluation. Gram-staining using the modified Brown and Brenn method (ScyTek, Logan, UT, USA) and periodic acid–Schiff (PAS; Abcam, Cambridge, UK) staining were performed, and examined histopathologically.

## 3. Results

### 3.1. Distribution of Corynebacterium spp. in Laboratory Mice

From 2015 to 2021, we isolated 165 strains of *Corynebacterium* spp. from laboratory mice in Korea and identified them using molecular methods, namely, 16S *rRNA* and *rpoB* gene sequence analysis. *Corynebacterium* infecting laboratory mice in Korea could be largely divided into four species: *C. amycolatum*, *C. bovis*, *C. lowii*, and *C. mastitidis* (Table 1).

Fourteen strains of *C. amycolatum* (8.5%) were isolated from mice, all of which showed 16S *rRNA* and *rpoB* gene sequences that were similar or identical to those of the *C. amycolatum* type strain. Strains that were not identical to the type strain differed by only 1 to 3 base pairs.

Forty-two strains of *C. bovis* (25.5%) were isolated, among which 38 showed 16S *rRNA* sequence identical to those of the *C. bovis* type strain; the remaining four strains had sequences identical to those of the *C. bovis* 99BR strain. The *rpoB* gene sequences of all 42 isolates of *C. bovis* were identical to the sequence of the *C. bovis*-type strain and were identical to each other.

Thirty-five strains of *C. lowii* (21.2%) were isolated and all isolates had 16S *rRNA* and *rpoB* gene sequence that were identical to those of the *C. lowii* type strain.

The most frequently isolated *Corynebacterium* was *C. mastitidis* (44.8%); 74 strains were separated, all of which had a 16S sequence that was identical to the type strain *C. mastitidis* S-8 or differed by 1–3 bases. The *rpoB* gene sequences of all 74 isolated strains of *C. mastitidis* were identical to that of the type strain.

No particular clinical symptoms were observed in any samples, except for the case of homozygous nude mice infected with *C. bovis*, some of which showed signs of hyperkeratosis, called “cornmeal coating” (Figure 1A). These mice were found to have white to yellow scaly flakes at the dorsum skin similar to that of a previous report [19]. Histopathological analyses demonstrated severe acanthosis, orthokeratotic hyperkeratosis, and infiltration of the dermis by lymphocytes, plasma cells, and neutrophils. PAS and Gram-staining revealed small rod- and cocci-shaped bacteria in the hyperkeratotic region (Figure 1B–D).

### 3.2. Biochemical Identification

Biochemical identification of *Corynebacterium* has historically been difficult because of their diversity and the fact that many biochemical tests needed for identification of human isolates of *Corynebacterium* spp. are not available in the API Coryne platform [21]. Therefore, in this study, we tested whether biochemical tests could accurately identify *Corynebacterium* spp. isolated from mice (Table 2).

Of the 14 strains of *C. amycolatum*, two were identified as *C. striatum/amycolatum*, one was identified as *Brevibacterium* spp., and 11 were unidentified. Thus, only two of 14 *C. amycolatum* strains were properly identified by biochemical methods, yielding a test sensitivity of 14.5% (2/14) (Table 2).

Of the 42 strains of *C. bovis*, 24 were correctly identified as *C. bovis*. The resulting test sensitivity was 57.1% (24/42) (Table 2). The remaining strains were identified as *C. urealyticum* (*n* = 5), *C. pseudotuberculosis* (*n* = 1), *Corynebacterium* group F1 (*n* = 1), *Rhodococcus* sp. (*n* = 1), *C. striatum*/*amycolatum* (*n* = 1), and unidentified (*n* = 9). 

Biochemical methods could not adequately identify any of the isolated *C. lowii* strains (Table 2). Of the 35 strains of *C. lowii*, 10 were identified as *C. bovis*, four were identified as *C. pseudodiphthericum*, two were identified as *C. striatum/amycolatum*, and 19 were unidentified.

*C. mastitidis* cannot be distinguished using the API Coryne system too; thus, as was the case for *C. lowii*, all strains were incorrectly identified by biochemical tests. The 74 strains of this species were characterized as follows: *C. urealyticum* (*n* = 25), *C. bovis* (*n* = 9), *C. pseudodiphtheriticum* (*n* = 5), *C. pseudotuberculosis* (*n* = 3), *C. propinquum* (*n* = 3), *C. kutscheri* (*n* = 2), *Brevibacterium* spp. (*n* = 1), and 26 were unidentified.

### 3.3. MALDI-TOF MS Identification

MALDI-TOF MS can be used to obtain protein fingerprints from whole bacterial cells that can be compared to a reference database using algorithms to rapidly identify bacteria [22]. To define the accuracy of MALDI-TOF MS identification of *Corynebacterium* in mice, we identified the 165 strains isolated using this method (Table 3).

Through MBT ver. 4.1.100, results obtained for the 14 strains of *C. amycolatum* were identical to those of molecular methods. Of the 42 strains of *C. bovis*, 41 were correctly identified as *C. bovis* and one was identified as *C. mastitidis*—a test sensitivity of 97.6%. None of the *C. lowii* strains identified by MALDI-TOF MS matched molecular biological identification results. As shown in Table 3, of the 35 strains of *C. lowii*, nine were misidentified as *C. mastitidis* and 26 were marked as “no identification”. Of the 74 strains of *C. mastitidis*, MALDI-TOF MS identified 72 correctly, with *C. bovis* (*n* = 1), and one sample as “no identification”, accounting for the two misidentifications. This result is 97.3% identical to that obtained using molecular methods. 

## 4. Discussion

*Corynebacterium* species are commonly found on the skin and mucous membranes of humans and animals, and are also found in various pathogenic substances [23]. *Corynebacterium* species have expanded rapidly. At present, a total of 133 species of *Corynebacterium* have validated names, of which 33 have been identified since 2016. This diversity and continued expansion of the genus makes it difficult to identify *Corynebacterium* species by physiological and biochemical methods.

At present, most identifications are achieved using molecular biological techniques, which are the methods of choice for identifying all *Corynebacterium.* However, these methods are expensive, time-consuming and require well-trained technicians, and thus are not generally used for diagnosing pathogenic bacterial infections. Recently, MALDI-TOF MS has been introduced for testing *Corynebacterium* isolated from humans and animals [24,25,26,27]. It has been shown to provide rapid, reliable identification results; however, some species were not properly identified [25]. Despite the use of MALDI-TOF MS for rapid identification of *Corynebacterium* strains from humans and commercial animals, this approach to identify *Corynebacterium* in laboratory mice has not been described well.

In this study, we isolated 165 *Corynebacterium* strains from laboratory mice in Korea and identified them using molecular and biochemical methods, as well as MALDI-TOF MS, and compared the results. As shown in Table 1, using a molecular approach, namely, sequencing of 16S *rRNA* and *rpoB* genes, we successfully identified all 165 isolates. Identification results obtained using 16S *rRNA* and *rpoB* gene sequence analysis were exactly the same at species level. Accordingly, for purposes of this study, these molecular identification results were used as reference standards for the other methods.

*C. amycolatum* is a common component of normal human skin flora [28], but has also been isolated in several human pathologies, including prosthetic joint infections [29], corneal ulcers [30], orbital implants [31], prosthetic valve endocarditis [32], and vaginitis [33]. However, there are no reports of *C. amycolatum* isolation or pathogenic effects in laboratory mice. Results of biochemical identification of *C. amycolatum* were only 14.3% identical to molecular results, whereas MALDI-TOF MS results were 100% identical. It should be noted that the impressive accuracy of the MALDI-TOF MS method relative to the biochemical identification method is based on a relatively small number of *C. amycolatum* strains. Thus, more tests should be performed to confirm these results, and additional studies on the differences between *C. amycolatum* isolated from mice and humans will be needed.

*C. bovis* has been reported to be pathogenic in immunocompromised mice and causes a scaly skin disease (Figure 1), with several such cases reported, including in Korea [7]. Only 57.1% of the biochemical identification results for *C. bovis* were the same as those obtained by the molecular biological method (Table 2). Biochemical identification results for *C. bovis* showed the highest homology with molecular methods among *Corynebacterium* spp. identified in this study (57.1%). MALDI-TOF MS results for this species—97.6% identical to molecular biological results—were much better (Table 3). Although MALDI-TOF MS analyses can be performed quickly, results are inaccurate for some strains, a problem that can be addressed by complementing MALDI-TOF MS analyses with molecular biological tests. Although *C. bovis* was reported to be pathogenic in mice, only some homozygote nude mice shown clinical signs in this study.

*C. lowii*, which was only recently distinguished from *C. mastitidis* [34], posed some problems for identification through biochemical methods and MALDI-TOF MS, as shown in Table 2 and Table 3. The current study provides the first evidence for infection and isolation of this species in mice, but we found no pathogenic effects in any of the infected mice. Although this bacteria was isolated from a patient undergoing ocular surgery [35], the pathogenicity of *C. lowii* in mice has not yet been elucidated and remains to be established.

*C. mastitidis*, originally found in the milk of sheep with subclinical mastitis and at the human ocular surface as a normal microflora [34,36], was the most common contaminant in mice. We isolated 74 strains and identified them using molecular, biochemical, and MALDI-TOF MS methods. Pathogenic effects of this bacterium in mice have not yet been well described, but Radaelii et al. reported isolation of *C. mastitidis* from two male mice with suppurative adenitis of preputial glands [37]. This bacterium has also been identified as an ocular commensal in C57BL/6 mice that drives the release of antimicrobials into the tears, an action that serves to protect the eye from pathogens such as *Candida albicans* and *Pseudomonas aeruginosa* [38]. In the current study, we found no specific clinical symptoms of *C. mastitidis* in mice. Both molecular biological methods and MALDI-TOF MS very effectively identified *C. mastitidis*, showing 97.3% similarity of results (Table 3). However, biochemical methods were completely unreliable, misidentifying all *C. mastitidis* strains (Table 2). This failure reflects the fact that API Coryne ver. 4.0, which currently can identify 33 *Corynebacterium* species, does not cover *C. mastitidis*.

The same problem plagued *C. kutscheri*, another well-known pathogenic *Corynebacterium* which can cause pseudotuberculosis [5]. *C. kutscheri* is usually a regular testing item in microbiological health monitoring and is very well controlled in Korean laboratory animal facilities [8]. Ultimately, we did not detect *C. kutscheri* infection in this study.

Previous biochemical tests on *C. diphtheria*, *C. ulcerans,* and *C. pseudotuberculosis*, which are well-known human pathogens, were 88.8% identical to identification results obtained using *rpoB* gene sequencing. However, concordance with *rpoB* gene sequencing results was even better using MALDI-TOF MS (99.1%) [24], reinforcing the view that MALDI-TOF MS provides an easy, fast, and reliable method for identifying *Corynebacterium* spp. Overall, MALDI-TOF MS results showed lower identity with molecular methods—77.0% in total—but still considerably better than biochemical identification results of *Corynebacterium* of mouse origin, which were only 15.8% identical to molecular methods in total. In a previous study on the low confidence in biochemical identification, Bernard [1] reported that this identification error is exacerbated in cases where the bacteria in question are metabolically inert, slow-growing, and express similar phenotypes. *Corynebacterium* spp. isolated from most mice grow slowly, consistent with this being a contributor to low accuracy, although a detailed investigation is warranted to confirm this. The reduced overall accuracy of MALDI-TOF MS was mainly attributable to incorrect identification of *C. lowii*, because, until recently, *C. lowii* was not taxonomically distinguished from *C. mastitidis*, so it has not yet been included in the producer’s database [30]. Accordingly, some strains were misidentified as *C. mastitidis*. Excluding the results for 35 strains of *C. lowii*, results of MALDI-TOF MS-based identification of *Corynebacterium* isolated from mice were 97.7% (127/130) consistent with those of molecular biology methods.

The commercially available MALDI Biotyper Compass database has limitations in identifying *Corynebacterium* strains from laboratory mice. MALDI-TOF MS has also shown misidentification or low identification scores in identifying *Corynebacterium* strains isolated from animals [39,40]. The addition of specific spectra to the entries, however, resulted in faster and more accurate identification. These findings suggest the importance of adjusting this method when using MALDI-TOF MS for the rapid identification of Corynebacterium strains isolated from laboratory mice.

## 5. Conclusions

Microbiological testing of laboratory animals is an important issue in the operation of laboratory animal facilities. As revealed in this study, different results may be obtained depending on the method used to identify *Corynebacterium* isolated from experimental mice, highlighting the importance of selecting an appropriate *Corynebacterium* identification method in obtaining accurate identification results. MALDI-TOF MS is an alternative testing method for rapid identification, but because this method has some limitations, it should be complemented by one or more different methods to allow more accurate *Corynebacterium* identification.

## Figures and Tables

**Figure 1 vetsci-09-00328-f001:**
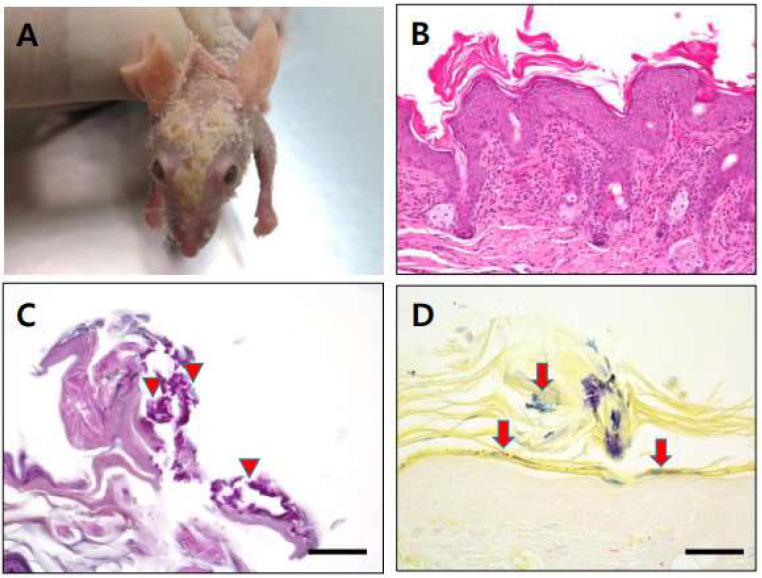
(**A**): Hyperkeratic skin lesion (cornmeal coating) was found in some *C. bovis*-infected immunodeficient nude mice. (**B**): Section of skin lesion (hematoxylin and eosin stain), severe acanthosis, orthokeratotic hyperkeratosis, infiltrate of lymphocyte with plasma cell, and neutrophils within dermis. (**C**): Small rod and cocci shape were found in hyperkeratic region after PAS staining (arrow head). (**D**): Gram-positive rod shape bacteria located in hyperkeratic area (arrow). Scale bars = 50 µm.

**Table 1 vetsci-09-00328-t001:** Distribution of *Corynebacterium* in experimental mice in Korea.

**Species**	**Strain**	**GenBank Accession No.**	**GenBank Accession No.**
		16S *rRNA* gene	*rpo**B* gene
*C. amycolatum* (14) ^a^	S160 ^T^ (14)	HE586271	CP069513
*C. bovis* (42)	Evans ^T^ (38)	NR_118465	CP066067
	99BR (4)	JX298786	
*C. lowii* (35)	R-50085 ^T^ (35)	NR_151864	KJ938692
*C. mastitidis* (74)	S-8 ^T^ (74)	Y09806	AY492281

^a^ = numbers in parentheses indicate the number of strains of bacteria isolated and identified. ^T^ = type strain.

**Table 2 vetsci-09-00328-t002:** Comparison of results of biochemical tests with molecular methods for identifying *Corynebacterium* spp.

	Molecular Method	API Coryne Ver. 4.0	Sensitivity
Species name	*C. amycolatum* (14)	Unidentified ^b^ (11)	2/14 (14.3%)
		*C. striatum/amycolatum*(2) ^a^	
		*Brevibacterium* sp.(1)	
	*C. bovis* (42)	*C. bovis* (24)	24/42 (57.1%)
		Unidentified ^b^ (9)	
		*C. urealyticum* (5)	
		*Corynebacterium* group F1 (1)	
		*C. pseudotuberculosis* (1)	
		*C. striatum/amycolatum* (1)	
		*Rhodococcus* sp. (1)	
	*C. lowii* (35)	Unidentified ^b^ (19)	0/35 (0%)
		*C. bovis* (10)	
		*C. pseudodiphthericum* (4)	
		*C. striatum/amycolatum* (2)	
	*C. mastitidis* (74)	Unidentified ^b^ (26)	0/74 (0%)
		*C. bovis* (9)	
		*C. pseudodiphtheriticum* (5)	
		*C. pseudotuberculosis* (3)	
		*C. propinquum* (3)	
		*C. kutscheri* (2)	
		*Brevibacterium* sp. (1)	
		*C. urealyticum* (25)	

^a^ = numbers in parentheses indicate the number of strains of bacteria isolated and identified. ^b^ = strains marked as “Unidentified” were defined as those for which the identification scores did not exceed the manufacturer’s guidelines.

**Table 3 vetsci-09-00328-t003:** Comparison of results of MALDI-TOF MS with molecular methods for identifying *Corynebacterium* spp.

	Molecular Method	MALDI-TOF	Sensitivity
Species name	*C. amycolatum* (14)	*C. amycolatum* (14) ^a^	14/14 (100%)
	*C. bovis* (42)	*C. bovis* (41)	41/42 (97.6%)
		*C. mastitidis* (1)	
	*C. lowii* (35)	no identification ^b^ (26)	0/35 (0%)
		*C. mastitidis* (9)	
	*C. mastitidis* (74)	*C. mastitidis* (72)	72/74 (97.3%)
		*C. bovis* (1)	
		no identification ^b^ (1)	

^a^ = numbers in parentheses indicate the number of strains of bacteria identified. ^b^ = if the score value did not note over 2.0 after ethanol/formic acid extraction, results were marked as “no identification”.

## Data Availability

Not applicable.

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
