# Peer review of "Distribution of Corynebacterium Species and Comparative Results of Diagnostic Methods for Identifying Corynebacterium in Experimental Mice in Korea"

_vetsci, 2022, doi:10.3390/vetsci9070328_

Round 1
Reviewer 1 Report
The authors showed an extent comparative among different methods to test the presence of Corynebacterium spp. in animal laboratory facilities. They found that traditional biochemical methods are not reliable for the less common species, highlighting the value of the molecular diagnosis and the importance of different methods at the same time.
The manuscript is well written and clearly exposes methodology and results. only minor changes should be done:
- Gram is more commonly used in capital
- 16S always in capital (lines 20, 67, 150, 243, 245)
- Gene names (rpoB -including the B- or rRNA for 16S) must be in italics, change along the manuscript.
- Both sp. and spp. not italicized (lines 182, 189, 267
- Sentence from lines 26-28 is confusing, rephrase.
- Corynebacterium in keywords
- C. mastiditis is n=75 or n=74?, clarify and change accordingly (lines 23, 153, 199...)
- Line 46: C. kutscheri
- Line 86: Trypticase Soy agar
- Line 92: add space between ver. and 4.0
- Line 143-144: what does similar means? Can you add a values as in line 155?
- Figure 1: please use same size in all the sections, aligned them and add scale bar in Figure B. Format of footnote must be standarize (A: - D: - B; - C. - chose a format); always space before parentheses; hyphen in Gram-positive; italics in C. bovis; cornmeal.
- Table 1, 2 and 3: add space betwwen the name and the parentheses. Can the font be changed to fit with the text?
- Line 279: manufaturer's
- Line 239: this approach is missing or is scarce
- Line 215: 26 got 'no identification' or had 'no identification'
- Line 313: items or item?
Author Response
Dear Reviewer 1, please see the attachment.

Reviewer 2 Report
General remarks:
The authors describe the identification of Corynebacterium-isolates from laboratory mice comparing molecular biological and biochemical methods as well as MALDI-TOF MS. The results are plausible and give a good idea of the capabilities and/or limitations of these methodological approaches.
Language needs improvement. In some cases unnecessarily long sentences are used.
Check references (e.g. 5 and 6 appear not to be mentioned in the text at all. 27 is referenced before 24, ref. 34 is first mentioned between ref. 14 and 15). Please correct and renumber.
The discussion is missing a major point that should be included prior to acceptance (see below).
Citation: Ki hoan Nam is missing from the author’s list
Line 20: “16s” should read “16S”.
Lines 21-23: I would suggest to list the isolated Corynebacterium sp. according to their percentage, not alphabetically.
Lines 23-24: Abbreviation is not needed in the abstract. If given anyway I’d suggest to change “matrix-assisted laser desorption/ionization-time of flight (MALDI-TOF) mass spectrometry” to “matrix-assisted laser desorption/ionization-time of flight mass spectrometry (MALDI-TOF MS)” as it is used in the following sentence.
Line 28: A non-identification is not a misidentification but no identification at all. The terms are not always interchangeable. The term “misidentification” is used very loosely throughout the text.
Line 28-29: 15% similarity to results of the molecular identification
Line 32-34: This sentence should be changed according to the edited discussion (see below)
Keywords: I suggest “MALDI-TOF” should be changed to “MALDI-TOF MS”
Line 41: As the number of C. species changes rapidly I would suggest to add “As of today” in the beginning of the sentence
Line 56: Specify “API” to “API-Coryne” as [8] does not use other APIs. Correct apostrophe position in “bioMe´rieux”.
Line 66: delete “recently” as Khamis et al. published their method in 2004/2005.
Line 87: add “aerobically”.
Line 107: add “on species level” after “properly obtained”.
Line 110: Specify the library version, e.g. add total number of entries.
The library version should also be indicated in chapter 3.3.
Lines 113-114: I suppose 16S rRNA sequences are partial too.
Lines 137-138: It would be good to give the number of laboratories the mice/isolates came from to give a better idea of the statistical relevance.
Lines 142-155: Genbank accession numbers are given in Table 1 so they could be left out here. If they are given in the text anyhow, add accession numbers for rpoB sequences (only accNo’s for 16 sRNA sequences are given).
Line 147: “DSM20582” is easier to read as “DSM 20582”.
Line 148: delete “\” in accession number JX29\8786.
Lines 152-155: why is no similarity of the rpoB sequences given as for the other species?
Line 154 & Table 1: Sequence NR_086376: NCBI says “This whole-genome-shotgun scaffold was removed because it has been superseded by a new assembly of the genome.” Please add new accession number.
Line 167ff: Use consistent format for “A: / B; / C. / D:”.
Some spaces are missing.
Line 178: Specify “API-Coryne” instead of “API”
Lines 186-187: The referenced Table 1 does not indicate to me that the sensitivity of the molecular identification was 57.1% (as well). If the test sensitivity of the molecular identification was only 57.1% how do you know that these strains are actually C. bovis?
Table 1: Adjust headings of columns 3 and 4.
Add spaces before parentheses.
“DSM 20582” is not a Genbank Accession no.
Does sequence CP066067 belong to both strains named in column 3?
Reword “a) In parentheses, it means the number of identified bacteria strains.” (see also lines 278 and 309)
Line 212: Again, if the test sensitivity of the molecular identification was only 97.6% how do you know that these strains are actually C. mastitidis?
Line 215: Here the use of “misidentified” instead of “identified as C. mastitidis” would be correct
Line 234: Add “and animals” and give references.
Table 2: Why does it say “Species name” in the column “Test method”? (same in Table 3)
API profile numbers?
Line 279: “manufature’s” is missing an r.
Line 316: This reference should be given after the previous sentence.
Line 328: “Bernald” should read “Bernard”.
Lines 345-348: If the limitations of the MALDI-TOF MS methods are mentioned (also lines 269-273), a short discussion of the possibilities of adding own reference spectra to the used database is neccessary, especially as the authors are using the Bruker machine that offers this option. This option is one of the major advantages of this method that has even lead to the discovery of new species in the past.
(see e.g. Badell et al., 2020 https://doi.org/10.1016/j.resmic.2020.02.003 or Rau et al., 2019 https://doi.org/10.1016/j.vetmic.2019.108399).
Author Response
Dear reviewer, Please see the attachmetnt

Reviewer 3 Report
Using different methods, Park et al. conducted an interesting study that evaluated the distribution of Corynebacterium species in mice, and they compared these methods in terms of efficacy, but also the disadvantages.
The introduction provides sufficient background to the subject. However, the introduction should be restructured. Data regarding Corynebacterium identification in humans should be presented separately from data regarding Corynebacterium identification in animals (mice).
We cannot cite and affirm that an article published in 2012 was published „more recently” (13).
Methods are included in the manuscript which is an advantage. Even though the data are provided in „supplementary materials”, it should be mentioned in the text the information regarding „ethical issues”.
Table 2 and Table 3 should be included in the results Section.
Discussions are clearly presented. I consider that the authors should include the conclusions of their study in a subsection at the end of the article. It was not possible to evaluate conclusions as a separate sub-chapter.
Author Response
Dear reviwer, please see the attachment.
